# RHAPSODY: A Dataset for Highlight Detection in Podcasts

**Younghan Park**[1]* **Anuj Diwan**[2]**, David Harwath**[2]† **Eunsol Choi**[3]†
[1]Yonsei University, [2]The University of Texas at Austin, [3]New York University
younghanpark@yonsei.ac.kr, {anuj.diwan, harwath}@utexas.edu, eunsol@nyu.edu

## Abstract

Podcasts have become daily companions for half a billion users. Given the enormous amount of podcast content available, highlights provide a valuable signal that helps viewers get the gist of an episode and decide if they want to invest in listening to it in its entirety. However, identifying highlights automatically is challenging due to the unstructured and long-form nature of the content. We introduce RHAPSODY, a dataset of 13K podcast episodes paired with segment-level highlight scores derived from YouTube's 'most replayed' feature. We frame the podcast highlight detection as a segment-level binary classification task. We explore various baseline approaches, including zero-shot prompting of language models and lightweight fine-tuned language models using segment-level classification heads. Our experimental results indicate that even state-of-the-art language models like GPT-4o and Gemini struggle with this task, while models fine-tuned with in-domain data significantly outperform their zero-shot performance. The fine-tuned model benefits from leveraging both speech signal features and transcripts. These findings highlight the challenges for fine-grained information access in long-form spoken media. We release our codes and dataset at https://github.com/younghanstark/rhapsody.

## 1 Introduction

Podcast consumption has grown significantly in recent years, prompting research into content-related tasks such as search (Tian et al., 2022), recommendation (Fan et al., 2023; Nadai et al., 2024), and summarization (Jones et al., 2020; Karlgren et al., 2021). However, identifying the most interesting parts (i.e., highlights) of podcast episodes remains relatively underexplored. This is an important problem, as the predicted highlights could help listeners quickly decide whether an episode is worth their time. Despite its various use cases, progress in this area has been limited due to several challenges: podcasts are often lengthy, and the most interesting parts of the podcast are subjective, making it challenging to gather coherent annotations.

In this paper, we present the first study on automatically locating highlights of podcast episodes. Our approach is inspired by the 'most replayed' feature of YouTube: a graph displaying frequently replayed parts of videos (referred to as the *replay graph*) (YouTube, 2022). This feature, which aggregates the views of thousands of users, opened doors for researchers to study video understanding tasks (Duico et al., 2023; Sul et al., 2023; Kim et al., 2025). Compared to such prior work which mainly consider visual features, we focus on the spoken content.

We first propose the podcast highlight detection task, and derive RHAPSODY,[1] a large-scale dataset (containing 13K podcast episodes from various domains), visualized in Figure 1. This task is long-context in nature because the average podcast episode in our dataset is about 30 minutes long and contains an average of 5K words. Each podcast is divided into 100 equal-length segments, of which a subset (with an average of 5 segments) is identified

---

*Work was partly done while at UT Austin.
†Equal advising.
[1]RHAPSODY: **R**eplay graph, **H**ighlights, **P**odcasts, **S**egment-level, **D**etection.

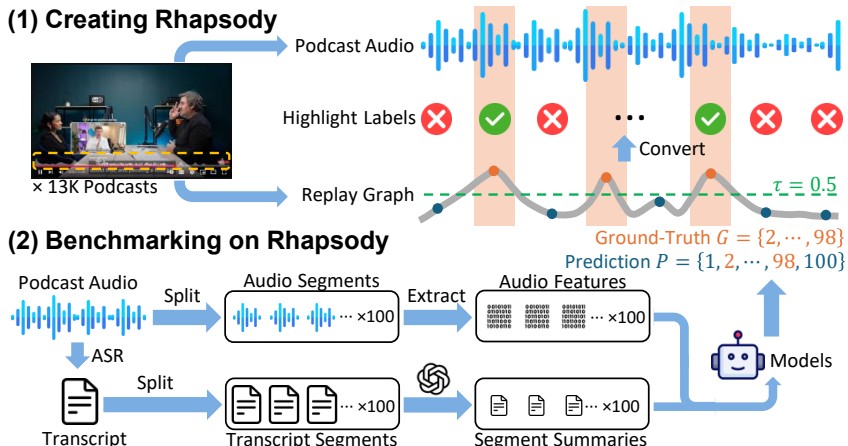

Figure 1: (Upper) RHAPSODY dataset derived from 13K podcast episodes from YouTube. The audio stream of an episode is split and paired with each data point in the replay graph of the video. (Lower) A model is asked to predict the indices of highlight segments, given the preprocessed textual and audio features of each episode segment. Ground-truth (GT) highlights are defined as indices of peaks in the replay graph (Section 2.2) and compared to the predicted indices for benchmarking.

as being the most-replayed highlights. We convert the replay graph into the subset of these most-replayed segments. Given input features from the podcast episode, such as episode transcript, episode title, or audio features, a model must identify this subset of most-replayed segments.

To tackle this new task, we propose a suite of baseline systems. We first present a text-based BM25 (Robertson & Zaragoza, 2009) approach, which treats highlight detection as the task of retrieving the segments whose transcript summaries (as summarized by an off-the-shelf LLM) have the highest BM25 score to the episode title. We next present zero-shot prompting results for two state-of-the-art large language models (OpenAI, 2024a; Google Deepmind, 2024). Lastly, we present learned models on this task, which uses hidden representation from a fine-tuned LLM and optionally audio features. This model outperforms GPT-4o's and Gemini's zero-shot performance by a large margin. Yet, the task remains very challenging, with a maximum hit rate of 49.0% with our best approach.

We sum up our contributions as follows:

- We formally describe the podcast highlight detection task, explain how supervision signals can be generated from replay graphs, and propose a set of automatic evaluation metrics for this task.
- We introduce RHAPSODY, a dataset of 13K podcast episodes from various domains, containing viewer replay scores for each segment.
- We benchmark the performance of various methods, including zero-shot LLM prompting and our fine-tuned language models, across text-only and text-and-audio settings.

We release all our data and code to support future research on podcast understanding. Given the limitations of current models, our benchmark can support rich future work, including methods that combine text and audio signals and methods that focus on the long-context ability of LLMs.

## 2 RHAPSODY: A Dataset for Highlight Detection in Podcasts

### 2.1 Task Formulation

YouTube provides replay graphs consisting of 100 time-aligned replay scores that display the most frequently replayed parts for most videos that have sufficiently high view counts

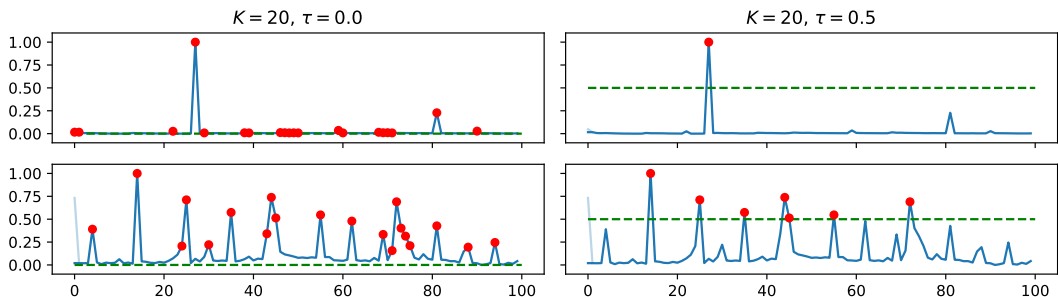

Figure 2: Comparing two settings for defining GT highlights indices $G$. (Left) Naively defining segments with top-$K$ replay scores as highlights. (Right) Introducing $\tau$, a hyperparameter to impose a minimum threshold (dotted green lines). The light blue lines at the beginning of the graphs represent the graph before bias correction (Appendix A.3).

(typically $\geq$ 50K views). We denote $A$ as the audio track of a podcast episode with a total duration $d$ and $M$ as the replay graph of $A$: $M = \{m_i\}_{i=1}^{100}$, where $m_i \in [0, 1]$. We divide the audio track $A$ into 100 equal-length segments $c_i$, $\frac{(i-1)d}{100}$ second to $\frac{id}{100}$ second, treating each $m_i$ as representing the replay score of its corresponding audio segment $c_i$. As described in Section 2.2, using the replay graph $M$, we derive a set of Ground-Truth (GT) highlight segment indices $G = \{g_j\}_{j=1}^{N_G}$ (where $g_j \in \{1, 2, \ldots, 100\}$) that correspond to the most-replayed segments in the podcast episode.

We propose a podcast highlight detection task where a model is provided input features derived from the podcast episode and is asked to predict a set of highlight indices $P = \{p_k\}_{k=1}^{N_P}$ (where $p_k \in \{1, 2, \ldots, 100\}$) which is compared to the GT highlight segment indices $G$. We consider both episode-level input features, such as the episode title, and segment-level input features, such as transcript summaries or audio features for each segment.

## 2.2 Data Collection Process

**Data Sources**  We scrape the list of most popular podcast creators from YouTube,[2] yielding 100 channel IDs. Using YouTube Data API v3,[3] we then collect metadata for these channels and their playlists. We discard playlists that (1) are not explicitly set as podcasts by their creators and (2) are not in English. We further filter out playlists that predominantly feature visual content, such as music videos, TV series reviews, or reaction videos. We also discard videos for which the replay graph is unavailable. Then, we limit the maximum duration of the episode to an hour to prevent a single data point in a replay graph corresponding to a long span of the audio. Moreover, we exclude episodes for which automatic transcription has more than 10 segments without any detected words. Details on filtering channels and playlists are in Appendix A.1. We download the audio track, replay graph, and YouTube video metadata for each podcast episode. The metadata includes episode-level features such as the title, creator-written description, video categories, and view statistics.

**Converting Replay Graph into Set of GT Highlight Indices**  Typically, only a small number of segments for each podcast have high replay scores. We convert the replay graph into a set of GT indices $G = \{g_j\}_{j=1}^{N_G}$ for evaluation. The simplest approach for defining $G$ is to set it as the indices of data points of the top-$K$ replay score (Duico et al., 2023; Sul et al., 2023). However, this approach causes false positive highlights when the replay graph is relatively flat. Instead, we introduce a hyperparameter $\tau$ representing a minimum replay score for being considered a highlight segment. Specifically, we label at most $K$ indices as GT among a set of candidate indices $\{i : m_i > \tau\}_{i=1}^{100}$. We opt to use $K = 20$ and $\tau = 0.5$ for

---

[2] https://www.youtube.com/podcasts/popularcreators
[3] https://developers.google.com/youtube/v3

| Category | # episodes | # channels | avg. duration (min) | # words (K) | # views (M) |
|---|---|---|---|---|---|
| Entertainment | 4,873 | 36 | 28.02 (15.49) | 4.95 (2.82) | 0.59 (1.32) |
| News & Politics | 4,154 | 16 | 24.65 (16.38) | 4.11 (2.81) | 0.47 (0.65) |
| Comedy | 632 | 15 | 25.50 (17.15) | 4.83 (3.33) | 0.29 (0.70) |
| Science & Tech. | 623 | 5 | 33.57 (18.80) | 5.52 (3.23) | 0.50 (0.96) |
| Education | 591 | 12 | 26.58 (17.34) | 4.37 (3.24) | 0.92 (1.79) |
| Sports | 496 | 11 | 29.58 (19.18) | 5.36 (3.66) | 0.27 (0.70) |
| People & Blogs | 474 | 24 | 33.37 (20.47) | 6.05 (3.87) | 0.68 (1.13) |
| Unknown | 1,317 | 66 | 27.04 (16.23) | 4.66 (2.90) | 0.63 (1.29) |
| All | 13,364 | 90 | 27.21 (16.70) | 4.71 (3.01) | 0.55 (1.13) |

Table 1: Statistics of podcast episodes in RHAPSODY. We report the number of podcast episodes and channels for each category, omitting categories with less than 100 episodes. We report the average duration of an episode, the number of words in the transcript, and the view count of the episode. We report the standard deviation in parentheses.

all experiments based on our manual data inspection. Figure 2 compares two settings for defining highlight segments. We provide more examples in Appendix A.2.

The replay graphs exhibit biases towards the first segment, likely as users frequently replay the beginning of the content without finishing it. In the original form, 75% of the episodes in the entire dataset have one of the GT highlights at the first segment. We correct this bias by setting the first value of the graph equal to the second one, preventing models from learning noisy patterns. Appendix A.3 provides more detailed explanation of bias correction.

**Dataset Statistics** Our dataset includes 13,364 examples, each corresponding to a single podcast episode. We randomly split our dataset into the train, validation, and test sets with a 70:10:20 ratio. Table 1 provides an overview of the statistics of podcast episodes.

## 3 Approaches for Predicting Podcast Highlights

We compare the performance of nine systems: random sampling and frequency-based baselines, a BM25-based method, three zero-shot prompting methods with GPT-4o and Gemini, and three fine-tuned LLMs—one with a segment-level binary classification head and the others additionally incorporating two types of audio features. We first describe feature extraction process from raw audio that will be used by systems.

### 3.1 Input Data Preprocessing

**Transcript Segmentation** To map the podcast transcript to the replay graph, which consists of 100 numbers, we split the transcript into 100 segments. We first run the podcast audio through WhisperX (Bain et al., 2023), a version of Whisper (Radford et al., 2022) that outputs a time-aligned ASR transcription providing word-level alignments of the form $\{(w_j, s_j, e_j)\}_{j=1}^{N_w}$, where $w_j$ represents the $j$th word, $s_j$ its start time, $e_j$ its end time, and $N_w$ the total number of words. Given this WhisperX output, we segment the transcript into 100 equal-duration intervals based on the episode segments $\{c_i\}_{i=1}^{100}$. A word $w_j$ is assigned to a segment if both $s_j$ and $e_j$ fall entirely within that segment's time span. If a word spans two segments, it is assigned to the segment with the greater overlap. We use `whisper-large-v2` for the transcription model and `WAV2VEC2_ASR_LARGE_LV60K_960H` (Baevski et al., 2020) for the alignment model.

**Summarization of Transcript Segments** We summarize the transcript segments using `gpt-4o-mini-2024-07-18` (OpenAI, 2024b) and use the summary output as input textual features to the model. We prompt LLM to process 10 consecutive segments at a time while providing the previous context to avoid providing an extremely long prompt. We provide more detailed information along with pseudocode in Appendix B.1.

**Audio Feature Extraction** To capture emotional characteristics in podcasts, we explore two types of audio features—dominance-valence-arousal (DVA) embeddings and HuBERT features. The DVA model represents emotions along three dimensions—dominance, valence, and arousal—providing a structured framework for analyzing affective states in speech (Russell & Mehrabian, 1977). To learn richer latent patterns, we leverage the hidden representation from the final transformer layer of an off-the-shelf DVA-based emotion recognition model (Wagner et al., 2023). For HuBERT features, we take the hidden representations from the 10th layer of the model, and average-pool them over time. A more detailed explanation of audio feature extraction can be found in Appendix B.2.

## 3.2 Models

**Random Sampling and Frequency-Based** As heuristic baselines, we consider random sampling and frequency-based baselines. The random sampling method samples $K$ unique elements from the population of 100 integers $\{1, 2, \cdots, 100\}$. The frequency-based method precomputes the occurrence count of each index across all GT highlight indices in the training set and selects the top-$K$ most frequent ones. In both methods, $K$ is sampled from the distribution of $|G|$, the number of GT indices, in the training set. All results from these baselines were averaged over 1,000 independent experiments.

**BM25** As a text-only baseline, we adapt BM25 for our task. We treat the summaries of each transcript segment as the corpus and the episode title as the query. We use 3.0 as the BM25 threshold, which is determined through manual tuning on the validation set.

**Zero-Shot Prompting with GPT-4o and Gemini** To test state-of-the-art LLMs on this long-context reasoning task, we consider three zero-shot settings: GPT-4o (text-only), Gemini (text-only), and Gemini (text + audio). For the GPT-4o (text-only) and Gemini (text-only) settings, we prompt `gpt-4o-2024-08-06` and `gemini-2.0-flash-001` with the title of the episode and transcript summaries for each segment, asking them to identify the segment indices that are most likely to be replayed. The model is instructed to rank answer segments in descending order of replay likelihood for proper computation of AP, and chain-of-thought prompting is applied in the prompt. For the Gemini (text + audio) setup, we also provide the full podcast audio. The text-only and the text + audio prompts can be found in Appendix C.2.

**Fine-tuned LLMs with Segment-Level Classification Heads** We fine-tune an LLM on RHAPSODY using parameter-efficient fine-tuning and replacing the default next-token prediction head with a custom segment-level binary classification head, shown in Figure 3. We develop three models, a **text-only** model that only utilizes textual features and two **text + audio** models that utilize both textual and audio features—DVA or HuBERT embeddings. The input to the text-only model is 100 transcript summaries, one for each of the 100 segments, with end-of-segment tokens appended after each segment. We extract one LLM hidden representation for each of the 100 end-of-segment tokens and apply a binary classification head, consisting of a linear layer and a sigmoid activation, to obtain 100 probability values, each estimating the likelihood that the corresponding segment is a highlight.

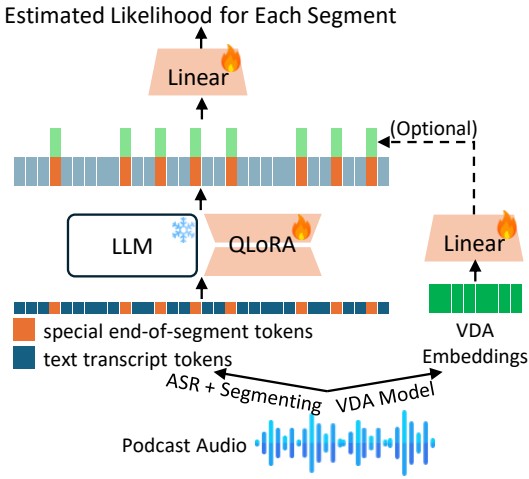

Figure 3: The illustration of proposed approach. Text and audio features are processed separately before being fused.

We freeze the base model, add QLoRA (Hu et al., 2022; Dettmers et al., 2023) adapters and optimize these adapters as well as the classification head and linear layer of audio features

| System | Hit Rate | Precision | Recall | F1 | AP |
|---|---|---|---|---|---|
| Random Sampling | 0.220 | 0.053 | 0.052 | 0.039 | 0.021 |
| Frequency-based | 0.224 | 0.072 | 0.064 | 0.051 | 0.031 |
| BM25 (text-only) | 0.216 | 0.051 | 0.060 | 0.04 | 0.024 |
| GPT-4o (text-only) | 0.265 | 0.088 | 0.053 | 0.057 | 0.032 |
| Gemini (text-only) | 0.327 | 0.082 | 0.074 | 0.065 | 0.037 |
| Gemini (text + audio) | 0.293 | 0.090 | 0.065 | 0.063 | 0.040 |
| Fine-tuned Llama (text-only) | 0.452 | 0.202 | 0.178 | 0.163 | 0.134 |
| Fine-tuned Llama (text + DVA) | 0.477 | 0.216 | 0.187 | 0.175 | 0.140 |
| Fine-tuned Llama (text + HuBERT) | **0.490** | **0.266** | **0.197** | **0.197** | **0.153** |

Table 2: Experimental results on test examples that have at least one ground-truth highlight ($|G| > 0$; 2,467 episodes). We bold the best-performing method and underline the second-best-performing method. We set all the metrics to be zero when there are no predicted indices ($|P| = 0$).

using binary cross-entropy loss. We use `Llama-3.2-1B-Instruct` (Meta, 2024) as the base model[4] and apply AdamW optimization with weight decay. We manually tuned hyperparameters on the validation split and selected the configuration that achieved the highest average performance across all evaluation metrics we report. As detailed in Appendix D, the hyperparameters we tune include the batch size, LoRA hyperparameters, learning rate, and prediction thresholds. The resulting configurations yielded approximately 5.6M, 6.7M, and 6.2M trainable parameters for the text-only, text + DVA, and text + HuBERT settings, respectively.

## 4 Experiments

### 4.1 Evaluation Metrics

We aim to measure the similarity of a sequence of predicted indices $P = \{p_k\}_{k=1}^{N_P}$ based on a sequence of GT indices $G = \{g_j\}_{j=1}^{N_G}$. Since this setting can be viewed as a document retrieval task where $P$ is a set of retrieved documents and $G$ is a set of relevant documents, we adopt five widely used evaluation metrics for document retrieval — hit rate, precision, recall, F1, and average precision (AP). When computing AP, we ensure $m_{g_i} \geq m_{g_j}$ is satisfied for all $1 \leq i < j \leq N_G$.

The number of model predictions heavily influences many evaluation metrics, such as precision and recall. Therefore, we tune the prediction thresholds of the BM25 and fine-tuned Llama models on the validation set, aiming to match the average number of predictions across all methods. We ensure that methods for which we cannot control the number of predictions, such as zero-shot prompting of GPT-4o and Gemini, do not produce excessively many predictions, to enable fair comparison with other models.

### 4.2 Results

We report the main experimental results in Table 2. Our experimental results show that even the best-performing model achieves only 0.266 precision and 0.197 F1 score, highlighting the difficulty of podcast highlight detection and the room for improvement beyond current approaches. Yet, this is a subjective task with derived gold labels, thus the upper-bound performance might not be perfect. We still anticipate a large room for improvement given the current performance level.

As expected, random sampling and frequency-based approaches performed the worst. Amongst the zero-shot prompting approaches, the Gemini model performed the best, while GPT-4o performed similarly to the frequency-based baseline, suggesting that it struggles

---

[4]We use `<|reserved_special_token_0|>` for end-of-segment token.

| Method | Ent. | News & Poli. | Comedy | Sci. & Tech. | Edu. | Sports | Ppl. & Blogs |
|---|---|---|---|---|---|---|---|
| Random Sampling | 0.240 | 0.157 | 0.238 | 0.227 | 0.227 | 0.257 | 0.211 |
| Frequency-based | 0.235 | 0.141 | 0.400 | 0.153 | 0.220 | 0.326 | 0.259 |
| BM25 (text-only) | 0.196 | 0.217 | 0.150 | 0.305 | 0.252 | 0.369 | 0.297 |
| GPT-4o (text-only) | 0.286 | 0.173 | 0.203 | 0.275 | 0.237 | 0.360 | 0.264 |
| Gemini (text-only) | 0.326 | 0.219 | 0.316 | 0.405 | 0.326 | 0.432 | 0.297 |
| Gemini (text + audio) | 0.315 | 0.185 | 0.233 | 0.351 | 0.319 | 0.360 | 0.253 |
| Fine-tuned Llama (text-only) | 0.489 | 0.384 | 0.714 | 0.313 | 0.326 | 0.586 | 0.527 |
| Fine-tuned Llama (text + DVA) | 0.541 | 0.403 | 0.714 | 0.374 | 0.341 | 0.441 | 0.516 |
| Fine-tuned Llama (text + HuBERT) | **0.642** | **0.427** | **0.820** | **0.420** | **0.444** | **0.649** | **0.637** |

Table 3: Average hit rate across subsets of podcast episodes grouped by genre. When multiple categories are listed in the video metadata, the first category is used as the genre.

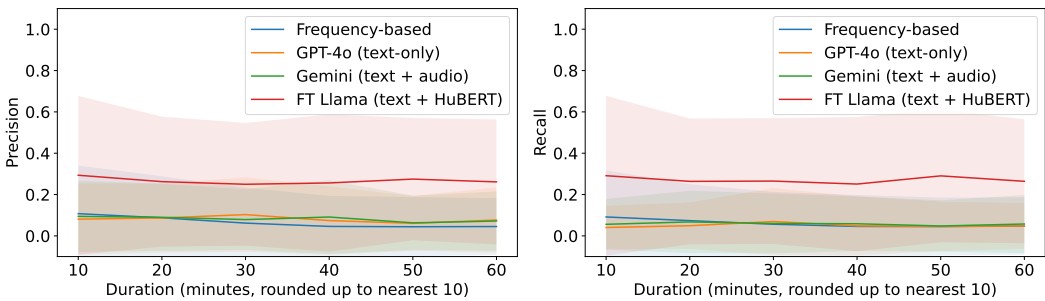

Figure 4: Precision (left) and recall (right) on test set, grouped by podcast duration. The shaded area represents ±1 standard deviation around the mean.

to infer highlight segments from textual content without explicit training. Finally, our fine-tuned LMs with a segment-level classification head trained on in-domain data, despite having only ∼ 6.2M trainable parameters, significantly outperform all baselines across all metrics. However, the absolute performance remains lacking, emphasizing the need for further advancements in this task.

**Effect of Audio Features**  Incorporating audio features alongside text leads to consistent performance improvements for the fine-tuned Llama model, suggesting that prosodic and acoustic cues provide valuable information for identifying highlights. Yet, the zero-shot Gemini result with audio does not consistently outperform the text-only Gemini result. This suggests that, while there are useful signals in the spoken content, fine-tuning with in-domain data might be needed to leverage such signals. Providing 100 segmented audio clips may contribute to improved zero-shot Gemini performance compared to providing the full episode as input.

**Performance on Different Genres and Duration Groups**  In Table 3, we present the average hit rate depending on podcast genres. Fine-tuned Llama model with HuBERT features performs the best, regardless of podcast genre. Notably, fine-tuned Llama models generally outperform other models in the 'Comedy' genre by a huge gap. It is also interesting that the fine-tuned Llama model without audio signals performs better than the one with DVA features in the 'Sports' genre. Additionally, LM-based methods don't perform as well in the 'News & Politics' genre as the other genres.

In Figure 4, we group podcast episodes by rounding up the durations to the nearest 10 minutes.[5] We do not observe a clear trend; most systems perform similarly regardless of podcast duration.

---

[5]Three LLM-based methods with a similar average number of predictions are plotted, along with the frequency-based baseline. We report the same set of methods also in Figure 5 and Figure 6.

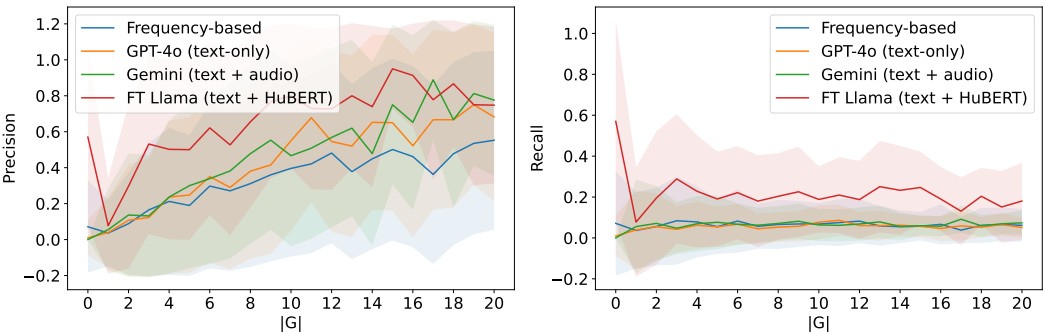

Figure 5: Precision (left) and recall (right) on test set, grouped by the number of GT high-lights. In $|G| = 0$ subset, we set both metrics to 1 if the model prediction is an empty set and otherwise 0. The shaded area represents $\pm 1$ standard deviation around the mean.

| System | # Highlights |
|---|---|
| Ground-Truth | 5.32 (4.51) |
| Random Sampling | 5.17 (4.41) |
| Frequency-based | 5.17 (4.41) |
| BM25 | 5.05 (5.36) |
| gpt-4o-2024-08-06 | 3.83 (1.16) |
| Gemini (text-only) | 6.03 (2.47) |
| Gemini (text + audio) | 4.32 (1.54) |
| FT Llama (text-only) | 3.94 (3.64) |
| FT Llama (text + DVA) | 3.96 (3.46) |
| FT Llama (text + HuBERT) | 3.54 (4.09) |

Table 4: The average number of GT and predicted highlights per episode for each method. We report the standard deviation in parentheses.

Figure 6: The average number of predicted highlights ($|P|$) for each subset grouped by the number of GT highlights ($|G|$). The shaded area represents $\pm 1$ standard deviation around the mean.

**Performance Across Varying Number of GT Highlights**   To analyze model performance based on the number of GT highlights per episode, we evaluate precision and recall across subsets of episodes grouped by their $|G|$. Figure 5 shows that the fine-tuned Llama model consistently outperforms all the other methods in terms of precision and recall in most subsets.

**Analysis on the Number of GT and Predicted Highlights**   In Table 4, we report the average number of predicted highlights ($|P|$) to ensure a fair evaluation of results. The baseline methods, on average, produce a higher number of predictions compared to all non-baseline methods. While zero-shot prompting with GPT-4o and Gemini (text + audio) produces a lower average $|P|$, their performance (as shown in Table 2) is superior to the baseline methods. This indicates that the actual performance gap between them is larger than what the reported numbers indicate. Furthermore, our proposed method achieves significantly better performance than all other approaches, including zero-shot GPT-4o, while maintaining a similar or lower average $|P|$. This supports our claim that our proposed approach is more effective.

Figure 6 further illustrates how each model adjusts its number of predicted highlights. Ideally, a model should predict the same number of highlights as the GT (dotted line). However, baseline models predict roughly the same number of highlights regardless of $|G|$. Notably, GPT-4o and Gemini also exhibit little variation in $|P|$ across different $|G|$ values, whereas our approach shows a clear upward trend, indicating a better alignment with the GT distribution.

## 5  Related Work

**Fine-Grained Podcast Information Access**   The podcast highlight detection task also falls under fine-grained information access tasks, as it focuses on identifying the most engaging segments within a podcast episode. Clifton et al. (2020) introduced 100,000 Podcasts, comprising nearly 60,000 hours of speech, along with their case study on spoken passage retrieval. Using this corpus, Jones et al. (2020) and Karlgren et al. (2021) further ran a track on podcast segment retrieval and received submissions from the public, while providing tasks, human annotations, and evaluation measures. Recently Ghazimatin et al. (2024) proposed PODTILE, which automatically generates chapters for a podcast episode, given its metadata and transcript. However, all of these works relied on human annotation, thus limited to smaller scale study.

**Video Highlight Detection**   Highlight detection is query-agnostic and inherently subjective task that requires multiple annotators. Due to the high cost of frame-wise or clip-wise annotation, earlier datasets are often small-scale (Sun et al., 2014; Song et al., 2015) or provide weak supervision (Gygli et al., 2016; Xiong et al., 2019). To address the need for large-scale datasets, Duico et al. (2023), Sul et al. (2023), and Kim et al. (2025) introduced the replay graphs for video highlight detection, showing that segments that are replayed many times align with human perception of highlights. Most works rely primarily on visual features (Lin et al., 2023; Sun et al., 2024; Yang et al., 2024). While some methods incorporate multimodal features (Liu et al., 2022; Zhou et al., 2024), non-visual features contribute little, and audio features used, such as PANN (Kong et al., 2019), do not capture spoken content. To our knowledge, no prior work has explored time-variant textual features, such as transcripts, for highlight detection in streaming media.

**Written and Spoken Passage Retrieval**   Traditional approaches rely on sparse representation such as BM25, while recent advancements leverage dense embeddings from neural models (Karpukhin et al., 2020; Wu et al., 2021; Yang et al., 2021) trained with contrastive learning objective. A few work also explores spoken passage retrieval (Lee et al., 2018; Lin et al., 2024). The podcast highlight detection task can be viewed as a form of passage retrieval, where the corpus consists of segmented podcast episodes, and the query is the entire episode itself.

## 6  Conclusion

In this work, we introduce the task of podcast highlight detection and present RHAPSODY, a dataset of 13K podcast episodes and binary highlight labels derived from YouTube's replay graph feature. We benchmark the performance of different approaches with five automatic evaluation metrics. Our benchmark results demonstrate that even state-of-the-art LLMs, such as GPT-4o, struggle with this task, performing on par with simple heuristics. In contrast, our fine-tuned 1B Llama model significantly outperforms all the other approaches. However, the absolute performance of the best model remains low, underscoring the challenges of detecting meaningful highlights in podcast episodes.

Future work can explore improved modeling and inference methods, such as incorporating rich audio or contextual cues to further enhance podcast highlight detection. Our system relies on separate components for transcription, summarization, audio feature extraction, and highlight prediction, which may introduce compounding errors. Joint optimization in an end-to-end framework or improved input processing may mitigate this. Additionally, the structure of replay graphs itself and adaptive strategies for different podcast genres are another promising direction.

## Acknowledgments

We thank all podcast creators on YouTube for creating interesting and valuable content. We also thank Youngjae Yu, Seungju Han, and Junhyeok Kim for their valuable input in the ideation of this work. Thanks to Xixi Hu for help with data collection.

## Ethics Statement

The dataset we introduce is created from podcast episodes available on YouTube, and we respect the rights of their respective creator. Our use of this data is strictly for academic research purposes. Following previous works utilizing publicly available videos from YouTube (Zellers et al., 2021; Han et al., 2023; Yang et al., 2023), we release only video IDs, replay graphs, and other processed features, while not releasing the original audio files. Additionally, we recognize that some podcasts in the dataset may contain sensitive content, including harmful, explicit, or discriminatory language. We encourage future research to consider ethical implications and mitigation of potential biases.

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

# A  Data Collection Details

## A.1  Data Filtering

**Filtering Channel IDs**  From the list of channel metadata gathered from the popular podcast creators webpage on YouTube, we exclude channels that satisfy at least one of the following conditions.

- Default language is set and is not English.
  ```
  "defaultLanguage" in channel_md["snippet"] and channel_md["snippet"]["
  defaultLanguage"] != "en"
  ```

In this step, we keep channels that do not have a default language set and they are further filtered in the following step in playlist-level if their content is not in English.

**Filtering Playlist IDs**  Among all publicly available playlists of the channels from the previous step, we only keep playlists that satisfy all of the following conditions:

- Creators have explicitly set them as podcasts.
  ```
  "podcastStatus" in playlist_md["status"] and playlist_md["status"]["
  podcastStatus"] == "enabled"
  ```
- Does not contain any filter words in its title.
  ```
  all([word not in playlist_md["snippet"]["title"] for word in ["music", "
  reaction", "review", "breakdown"]])
  ```
- The playlist is assumed to be in English.
  ```
  isEnglish(' '.join([playlist_md["snippet"]["title"], playlist_md["
  snippet"]["channelTitle"], playlist_md["snippet"]["description"]]))
  ```

To filter non-English playlists out, we first concatenate the playlist title, channel title, and playlist description, using a separator as a space—an *identifier string*. Next, we utilize Lingua[6], a lightweight library for language identification, to analyze the identifier string. Playlists are assumed to be in English if (1) the highest-confident language is English or (2) has a confidence score lower than 75%. The second condition is added based on the authors' manual inspection of filtering results to avoid false negatives.

---

[6]https://github.com/pemistahl/lingua-py

## A.2 Converting Replay Graph into Set of Highlight Indices

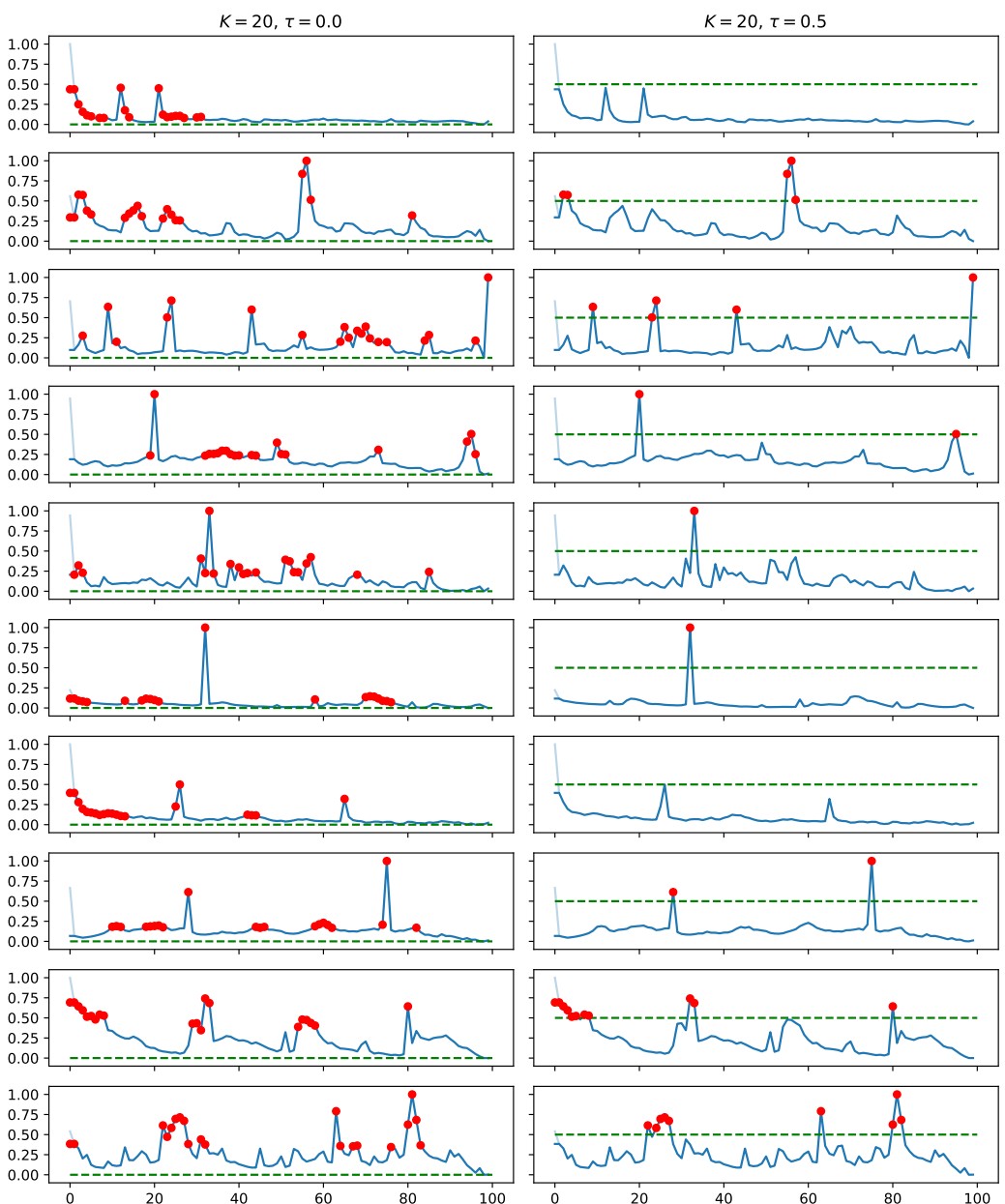

Figure 7: More examples comparing two different hyperparameter settings for converting a replay graph into a set of GT highlights.

### A.3 Bias Correction in Replay Graphs

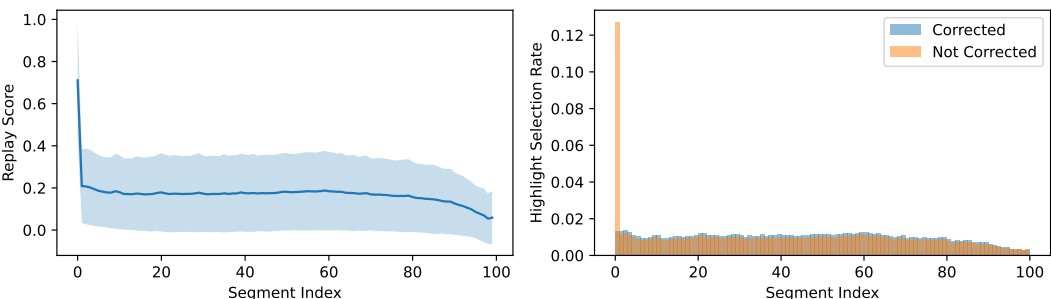

Figure 8: (Left) The average replay graph over all replay graphs in RHAPSODY. The shaded area represents the standard deviation at each index. (Right) The highlight selection rate for each segment index before and after bias correction.

As shown in the left graph in Figure 8, the replay graphs internally have biases, particularly at the first segment. Specifically, the first value (0.711) is significantly higher than the rest, which ranges from 0.054 to 0.208. This pattern likely arises because users frequently replay the beginning of the content after missing the start, rather than due to the segment's inherent saliency. Since this spike reflects an artifact of the user's watching behavior rather than meaningful engagement, we correct it by setting the first value equal to the second value, i.e. `graph[0] = graph[1]`. As shown in the right plot of Figure 8, this correction has little effect on the distribution of highlights in segments other than the first. The average number of highlights per episode $|G|$ decreases slightly from about 5.86 to 5.20.

## B Input Data Preprocessing Details

### B.1 Transcript Segment Summarization

Algorithm 1 describes our multi-stage summarization pipeline, which aims to improve the quality of summaries and coherence across transcript segment summaries. Given 100 transcript segments, the algorithm processes them in batches of 10. At each stage, the LLM generates individual summaries for the batch and a combined summary incorporating contextual information from the previous batch. The first stage has no prior context, so it starts with an empty summary.

---

**Algorithm 1** Multi-Stage Summarization

---

1: **Input:** Transcript segments $\{t_i\}_{i=1}^{100}$
2: **Initialize:** $S_0 \leftarrow$ "None.", $\{\hat{t}_i\}_{i=1}^{100} \leftarrow$ uninitialized array of size 100
3: **for** $k = 1$ to 10 **do**
4: $\quad B_k \leftarrow \{t_i\}_{i=10(k-1)+1}^{10(k-1)+10}$
5: $\quad \hat{B}_k, S_k \leftarrow$ LLM$(B_k, S_{k-1})$
6: $\quad$ Store $\hat{B}_k$ into corresponding indices in $\{\hat{t}_i\}_{i=1}^{100}$
7: **end for**
8: **Output:** Summarized transcript segments $\{\hat{t}_i\}_{i=1}^{100}$, summaries for each batch $\{S_k\}_{k=1}^{10}$

---

The full prompt used can be found in Appendix C.1.

## B.2 Audio Feature Extraction

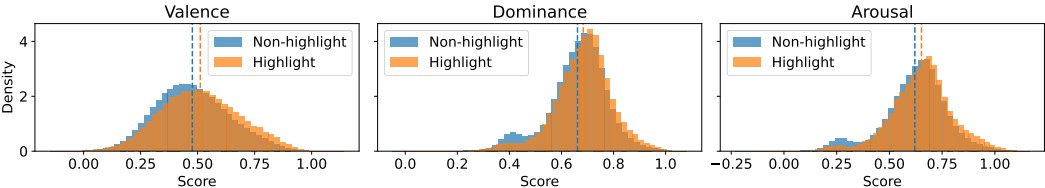

Figure 9: Valence, dominance, arousal score density distributions for highlight and non-highlight segments. Dotted vertical lines represent the mean of the distribution.

We use `audeering/wav2vec2-large-robust-12-ft-emotion-msp-dim` for DVA feature extraction, and `facebook/hubert-base-ls960` for HuBERT feature extraction. The model produces a 1024-dimensional and 512-dimensional embedding from an input audio file, respectively. To obtain 100 embeddings per episode, we first load the raw audio at a 16,000Hz sample rate, split it into 100 segments, and process each through the model.

Specifically, the DVA embeddings can be further used to predict valence, dominance, and arousal scores with their pretrained regression head. As shown in Figure 9, there is a slight but obvious difference in all dimensions. For example, the average score is slightly higher for highlighted segments across all dimensions.

## C  Used Prompts

We denote the variables in the template with {`teletype font and curly brackets`}.

### C.1  Multi-Stage Summarization of Transcript Segments

---

**SYSTEM** You are an expert in summarizing podcast transcripts into concise summaries. Your tasks are:
1. Summarize each provided segment, ensuring a coherent logical flow.
2. Generate a combined summary for all the provided segments, focusing on conveying the overall context efficiently without unnecessary detail.
3. If a previous summary is provided, ensure your summaries align with it.
4. Focus on providing a concise overview, omitting minor details when necessary to ensure clarity.
5. Produce output in the following format:
- Segment X: [Summary of Segment X]
- Segment Y: [Summary of Segment Y]
- ...
- Segment Z: [Summary of Segment Z]
- Combined Summary: [Summary of all segments combined]

---

**USER** Summarize the following podcast transcript into individual segment summaries and a combined summary for all the provided segments.
Podcast Title: {`title`}
Previous Summary (if any): {`combined summary of the previous batch`}
Segments:
- Segment 0: {`transcript of segment 1`}
{`transcript of segment 2 to 9 in the same format`}
- Segment 9: {`transcript of segment 10`}

---

Table 5: Prompt template for summarizing transcript segments.

## C.2 Zero-Shot Prompting with GPT-4o and Gemini

---

**SYSTEM** You are an assistant trained to analyze summaries of podcast segments and predict listener behavior. Your task is to identify segments that are most likely to be replayed, rank them by likelihood, and provide concise and specific rationales. Follow these guidelines when answering:
1. Rank segments in descending order of replay likelihood.
2. Include a segment only if there is a highly obvious reason it might be replayed, excluding uncertain segments.
3. If no segments meet the criteria, leave the 'Answer' item empty.
4. Respond in the following format:
- Segment X: (one-line rationale for why it is replayed)
- Segment Y: (one-line rationale for why it is replayed)
...
- Segment Z: (one-line rationale for why it is replayed)
- Answer: X, Y, ..., Z

---

**USER** Analyze the following podcast segments:
- Title: {title}
- Segment 0: {summary of transcript segment 1}
- Segment 1: {summary of transcript segment 2}
{summary of transcript segment 3 to 98 in the same format}
- Segment 98: {summary of transcript segment 99}
- Segment 99: {summary of transcript segment 100}
Which segments are most likely to be replayed?

---

Table 6: Text-only prompt template for identifying highlight segments.

---

**SYSTEM** You are an assistant trained to analyze summaries of podcast segments and predict listener behavior. You will be provided the summaries of podcast segments as well as the original podcast audio. Your task is to identify segments that are most likely to be replayed, rank them by likelihood, and provide concise and specific rationales. You should use both the provided summary text and the audio to make your decision, paying attention to the content of the text and audio as well as prosodic and emotional cues from the audio. Follow these guidelines when answering:
1. Rank segments in descending order of replay likelihood.
2. Include a segment only if there is a highly obvious reason it might be replayed, excluding uncertain segments.
3. If no segments meet the criteria, leave the 'Answer' item empty.
4. Respond in the following format:
- Segment X: (one-line rationale for why it is replayed)
- Segment Y: (one-line rationale for why it is replayed)
...
- Segment Z: (one-line rationale for why it is replayed)
- Answer: X, Y, ..., Z

---

**USER** {podcast audio} Analyze the following podcast segments:
- Title: {title}
- Segment 0: {summary of transcript segment 1}
- Segment 1: {summary of transcript segment 2}
{summary of transcript segment 3 to 98 in the same format}
- Segment 98: {summary of transcript segment 99}
- Segment 99: {summary of transcript segment 100}
Which segments are most likely to be replayed?

---

Table 7: Text + audio prompt template for identifying highlight segments.

## D    Model Implementation Details

**Zero-shot prompting with LLMs** For both `gpt-4o-2024-08-06` and `gemini-2.0-flash-001`, we use default hyperparameter settings. The string following `Answer:` in the LLM response is parsed and converted into a set of predicted indices.

**Fine-Tuned LLM with Token-Selective Classification Head**    The model is trained using a batch size of 1 and a gradient accumulation step of 8. QLoRA is applied with a dropout rate of 0.3, a rank of 8, and an alpha value of 16. The optimizer used is AdamW with a weight decay of 0.01 and a learning rate of 0.0003. We train the model with 4 NVIDIA RTX8000 GPUs. The model outputs the indices in the graph above a certain threshold as predictions. We use the prediction threshold of 0.3, 0.425, and 0.375 for text-only, text + DVA, and text + HuBERT settings, respectively. These values are determined by the authors' manual tuning on the validation set. We take checkpoints with the best average metrics on the validation set.

## E    Attribution of Icons Used

All the icons used in this paper are from https://www.flaticon.com/.

| Location | Icon | Author |
|---|---|---|
| Figure 1 | Wave Sound | Freepik |
| | Document | Freepik |
| | Check | hqrloveq |
| | Delete | hqrloveq |
| | Code | pikepicture |
| | Robot | Hilmy Abiyyu A. |
| Figure 3 | Freezing | Freepik |
| | Fire | Bahu Icons |

Table 8: Author information for all icons used.

