# OpenReview forum: "Rhapsody: A Dataset for Highlight Detection in Podcasts"
_colmweb.org/COLM/2025/Conference — COLM 2025_

### Official Review · Reviewer_HBck · 2025-05-12

**Rating:** 6
**Confidence:** 5
**Ethics Flag:** 2

**Summary:**

This paper introduces RHAPSODY, a dataset of 13K podcast episodes for highlight detection, where "highlights" are derived from YTB's "most replayed" feature. The authors formulate highlight detection as a segment-level binary classification task and benchmark various commonly used approaches.

**Questions To Authors:**

1. Why were emotion embeddings used for audio features? What motivated this choice over other audio representations?
2. Can you provide qualitative examples of segments that are incorrectly classified as highlights to better understand model behavior?
3. How do results vary across different podcast genres?
4. What is the human performance on this task?

**Reasons To Accept:**

1. This paper introduces the first large-scale dataset for podcast highlight detection; The use of YouTube's replay graph as a proxy for highlights and does not require manual labeling.
2. The authors evaluate a well-designed range of approaches from simple heuristics to retrieval methods (BM25) to state-of-the-art LLMs (GPT-4o, Gemini) in both zero-shot and fine-tuned settings. This provides a thorough baseline for future works.

**Reasons To Reject:**

1. While the paper demonstrates that audio features improve performance, it lacks sufficient justifications: only one type of audio embedding is explored (VDA emotion embeddings from wav2vec2, Section 3.1 and Appendix B.2), without justification for this specific choice. No ablation studies on alternative audio representations (e.g., raw spectrograms, CLAP embeddings, or other pre-trained audio models)

2. The paper's writing could be improved: (1) The contribution list (lines 53-60) should follow a more logical order: (a) problem formulation, (b) dataset creation, (c) benchmark systems; (2) The motivation for the task is underdeveloped in the introduction; (3) using "Feature-agnostic" (line 108) to categorize different methods is misleading, (4) System descriptions are scattered across Sections 3.1 and 3.2 (are data preprocessed in section 3.1 used in all of the systems in sec3.2? if not, that step is not data preprocessing but one steo of the system); (5) The distinction between "comparison systems" and experimental setup is unclear, from what I saw, the baselines are just random/frequency based methods, and the rest are all proposed systems.

3. The evaluation metrics lack clear definitions and explanations; The discussions on "Analysis on the Number of GT and Predicted Highlights" (lines 232-247) and "Performance Across Varying Number of GT Highlights" (lines 248-252) are particularly confusing: (1) Figure 4's purpose and interpretation are unclear - why is it important that models predict similar numbers regardless of |G|? (2) The transition between discussing quantity (Table 3, Figure 4) and performance metrics (Figure 5) lacks coherence. These analyses feel disconnected from the main narrative about model performance

---

> ### Author Response · Authors · 2025-06-03
> **Author Response for Reviewer HBck (1/2)**
>
> **[R1] While the paper demonstrates that audio features improve performance, it lacks sufficient justification.**
>
> To strengthen our argument that audio features improve performance, we additionally explore HuBERT embedding (https://huggingface.co/facebook/hubert-base-ls960) [1]. Specifically, we take the embeddings at the 10th layer and average pool them over time. Please refer to the performance difference in the following table:
> | System               | Hit       | Prec.     | Recall    | F1        | AP        |
> |----------------------|-----------|-----------|-----------|-----------|-----------|
> | Ours (text-only)     |     0.452 |     0.202 |     0.178 |     0.163 |     0.134 |
> | Ours (text + VDA)    |     0.477 |     0.216 |     0.187 |     0.175 |     0.140 |
> | Ours (text + HuBERT) | **0.490** | **0.266** | **0.197** | **0.197** | **0.153** |
>
> As shown in this table, our approach benefits from both VDA and HuBERT embeddings. Notably, the system with HuBERT embedding performs the best among all the proposed systems, while having a smaller number of predicted highlights than others (3.73). We will add this experimental result to the updated manuscript. Thank you for the suggestion!
>
> **[R2] The paper's writing could be improved.**
>
> Thank you for your suggestions, we will make edits in the draft accordingly:
> - (1) We will switch the order of the first and second items of the contributions list.
> - (2) We will strengthen the motivation by adding discussions on use cases.
> - (3) We will not use the term 'feature-agnostic' and refer to the methods directly.
> - (4) The processing steps in Section 3.1 apply to all systems in Section 3.2, and we thus separate the subsections in the current way.
> - (5) We will make the distinction between baselines (random/frequency-based) and non-baselines clearer.
>
> **[R3a] Figure 4's purpose and interpretation are unclear - why is it important that models predict similar numbers regardless of |G|?**
>
> We pointed out the fact that baseline models predict roughly the same number of highlights regardless of |G| to emphasize the clear upward trend, which means the better alignment of our proposed approach.
>
> **[R3b] The transition between discussing quantity (Table 3, Figure 4) and performance metrics (Figure 5) lacks coherence.**
>
> We think your suggestion is reasonable. We discuss performance metrics first (Table 2), then quantity metrics (Table 3, Figure 4), and then performance metrics again (Figure 5). We will switch the order of the last two items to have better coherence.
>
> [1] Hsu, Wei-Ning, et al. HuBERT: Self-Supervised Speech Representation Learning by Masked Prediction of Hidden Units. IEEE/ACM Transactions on Audio, Speech, and Language Processing 2021

---

> > ### Author Response · Authors · 2025-06-03
> > **Author Response for Reviewer HBck (2/2)**
> >
> > **[Q1] Why were emotion embeddings used for audio features over other audio representations?**
> >
> > Emotion embeddings are intuitively a good choice because podcast highlights are often related to emotional peaks (tension, excitement, etc.). The specific embeddings we used were finetuned on dominance, valence, and arousal data; dominance/arousal often indicates excitement, and valence indicates positive/negative emotion, making them a compact representational choice for our task.
> >
> > **[Q2] Can you provide qualitative examples of segments that are incorrectly classified as highlights?**
> >
> > Thanks for the suggestion, we will add another section in the appendix and provide some qualitative examples in the camera-ready version of the paper.
> >
> > **[Q3] How do results vary across different podcast genres?**
> >
> > For your better understanding of the result, we report the hit rate of the models depending on the podcast genre:
> > |Method|Entertainment|News & Politics|Comedy|Science & Tech.|Education|Sports|People & Blogs|
> > |---|---|---|---|---|---|---|---|
> > |Random Sampling|0.24|0.16|0.24|0.23|0.23|0.26|0.21|
> > |Frequency-based|0.24|0.14|0.40|0.15|0.22|0.33|0.26|
> > |BM25 (text-only)|0.20|0.22|0.15|0.31|0.25|0.37|0.30|
> > |GPT-4o (text-only)|0.29|0.17|0.20|0.27|0.24|0.36|0.26|
> > |Gemini (text-only)|0.33|0.22|0.32|**0.40**|0.33|0.43|0.30|
> > |Gemini (text + audio)|0.32|0.18|0.23|0.35|0.32|0.36|0.25|
> > |Finetuned Llama (text-only)|0.49|0.38|**0.71**|0.31|0.33|**0.59**|**0.53**|
> > |Finetuned Llama (text + audio)|**0.54**|**0.40**|**0.71**|0.37|**0.34**|0.44|0.52|
> >
> > Finetuned Llama models perform the best, except for the science & technology genre. Notably, they outperform other models in the comedy genre by a huge gap. It is also interesting that the finetuned Llama model without audio signals performs better than the one with audio signals in the sports genre. In general, LM-based methods don’t perform as well in the news & politics genre as the other genres.
> >
> > As this analysis provides interesting findings, we will add this table and the discussion in the camera-ready version of the paper.
> >
> > **[Q4] What is the human performance on this task?**
> >
> > Considering that the average duration of the podcasts in our dataset is ~30 minutes and the average number of words in the transcripts is 5K, we were unable to measure reliable human performance on the task. Instead, we would like to point you towards a user study about the alignment between human perception and YouTube’s replay peaks in the previous literature [1]. To quote their results (Table 5 in [1]), the annotators agreed with 64.8% of the labels.
> >
> > [1] Sul, Jinhwan, et al. Mr. HiSum: A Large-scale Dataset for Video Highlight Detection and Summarization. NeurIPS 2023

---

> ### Comment · Reviewer_HBck · 2025-06-08
>
> The response looks good, thanks for adding the experiment. Looking forward to the revised version.

---

### Official Review · Reviewer_ZtNz · 2025-05-13

**Rating:** 6
**Confidence:** 5
**Ethics Flag:** 1

**Summary:**

This paper explores the challenge of automatically identifying highlights in podcasts. It introduces RHAPSODY, a dataset of 13,000 podcast episodes. The authors frame highlight detection as a binary classification task and use a set of models including zero-shot prompting and fine-tuned LLMs for this task. Experimental results show that even advanced models like GPT-4o and Gemini do not perform well in highlight detection.

**Reasons To Accept:**

1- The introduction of the RHAPSODY dataset is a valuable contribution. There are a few podcast datasets available in the community and this dataset can extend the research in this very important field.

2- The task of highlight detection is also new and well-motivated, providing a benchmark that the community can build upon.

3- By combining speech signal features and transcripts, the paper provides evidence that multimodal representations significantly improve performance. This insight could inform future research in multimodal and spoken content understanding.

**Reasons To Reject:**

1. While highlight detection in video content often aligns with visually or emotionally engaging moments, podcast highlights are inherently semantic and contextual. They typically reflect importance, novelty, or narrative turning points, I believe these are qualities that are better captured by extractive summarization approaches like TextRank. This distinction is not explored in the paper, which can be an interesting addition. Additionally, different podcast formats (e.g., interviews, solo speaker, storytelling) may require different definitions and strategies for highlight detection. The authors should acknowledge this variation and discuss its implications for model training and evaluation.
2. Podcasts span a wide range of genres, each with a unique structure and style. These differences can influence highlight characteristics. However, the paper does not evaluate how genre affects model performance, which limits our understanding of the model’s generalizability and robustness.
3. Many podcasts are published with creator-written descriptions or summaries, which are used in other benchmark datasets (e.g., Spotify) as ground-truth references [1]. The paper does not specify whether these summaries were collected, and/or does not justify the decision to disregard these human-authored annotations.
4. Podcasts often contain ads, sponsor messages, intros, or other repetitive boilerplate sections that are not semantically meaningful [2]. The paper does not explain whether such segments were removed or how the models are expected to handle them. These segments could add noise to both the training and model predictions.
5. The reported performance is relatively low, even for large models like GPT-4o and Gemini, which raises concerns about the task formulation and the reliability of the evaluation method. While the finetuned models perform better, they still struggle with precision. This may indicate that the models are not capturing what human listeners actually perceive as highlights. Since the evaluation is completely automatic and based on replay-based supervision, it remains unclear whether replay peaks truly align with human judgment. Incorporating human evaluation of predicted highlights would strengthen the paper and help validate the task formulation and model outputs.


[1] Ann Clifton, Sravana Reddy, Yongze Yu, Aasish Pappu, Rezvaneh Rezapour, Hamed Bonab, Maria Eskevich, Gareth Jones, Jussi Karlgren, Ben Carterette, and Rosie Jones. (2020). 100,000 Podcasts: A Spoken English Document Corpus. In Proceedings of the 28th International Conference on Computational Linguistics, pages 5903–5917, Barcelona, Spain (Online). International Committee on Computational Linguistics.

[2] Reddy, S., Yu, Y., Pappu, A., Sivaraman, A., Rezapour, R., & Jones, R. (2021, April). Detecting Extraneous Content in Podcasts. In Proceedings of the 16th Conference of the European Chapter of the Association for Computational Linguistics: Main Volume (pp. 1166-1173).

---

> ### Author Response · Authors · 2025-06-03
> **Author Response for Reviewer ZtNz**
>
> **[R1a] Extractive summarization approach**
>
> We agree that extractive summarization methods could generate better text features for detecting highlights. However, we used abstractive summarization rather than extractive summarization due to the noise of raw transcripts. For instance, they don’t provide any delimitation between the interlocutors or the time information between utterances. We used LLM-based abstractive summarization to clean the noise and extract important information at the same time. In summary, the format of the data makes the use of an extractive approach challenging. We will add this discussion to the final paper.
>
> **[R1b] Different podcast formats (e.g., interviews, solo speaker, storytelling) may require different definitions and strategies for highlight detection.**
>
> As you mentioned, there is a variation in the format of podcasts, and the best strategy for detecting highlights might be different depending on the format of the podcasts. Future work could explore dynamic inference strategies, such as prompt tuning, to better adapt to the podcast format. We will mention your suggestion as one of the future directions in the conclusion section.
>
> **[R2] The paper does not evaluate how genre affects model performance.**
>
> For your better understanding of the result, we report the hit rate of the models depending on the podcast genre:
> |Method|Entertainment|News & Politics|Comedy|Science & Tech.|Education|Sports|People & Blogs|
> |---|---:|---:|---:|---:|---:|---:|---:|
> |Random Sampling|0.24|0.16|0.24|0.23|0.23|0.26|0.21|
> |Frequency-based|0.24|0.14|0.40|0.15|0.22|0.33|0.26|
> |BM25 (text-only)|0.20|0.22|0.15|0.31|0.25|0.37|0.30|
> |GPT-4o (text-only)|0.29|0.17|0.20|0.27|0.24|0.36|0.26|
> |Gemini (text-only)|0.33|0.22|0.32|**0.40**|0.33|0.43|0.30|
> |Gemini (text + audio)|0.32|0.18|0.23|0.35|0.32|0.36|0.25|
> |Finetuned Llama (text-only)|0.49|0.38|**0.71**|0.31|0.33|**0.59**|**0.53**|
> |Finetuned Llama (text + audio)|**0.54**|**0.40**|**0.71**|0.37|**0.34**|0.44|0.52|
>
> Finetuned Llama models perform the best, except for the science & technology genre. Notably, they outperform other models in the comedy genre by a huge gap. It is also interesting that the finetuned Llama model without audio signals performs better than the one with audio signals in the sports genre. In general, LM-based methods don’t perform as well in the news & politics genre as the other genres.
>
> As this analysis provides interesting findings, we will add this table and the discussion in the camera-ready version of the paper.
>
> **[R3] Collection and use of creator-written descriptions or summaries**
>
> We collect and plan to release YouTube metadata of podcasts in the dataset, and the metadata includes the video’s description on YouTube. We have considered using the video description, but decided not to use them since we noticed that descriptions on YouTube often include unrelated content, such as advertisements and links to the creator’s SNS, unlike descriptions in the Spotify podcast dataset. That being said, we agree that it is unclear in our writing what data was collected for each video and what items are available in the metadata. We will make edits in the final draft to make it clearer.
>
> **[R4] Handling of non-content segments (e.g., ads, intros, boilerplate)**
>
> We agree that there might exist extraneous content in podcasts. However, we believe that LMs should ideally consider those factors and correctly distinguish between extraneous and non-extraneous content. Furthermore, our approach is capable of handling those segments, as LLMs are asked to summarize raw transcripts while focusing on conveying the overall context without unnecessary detail (Section 3.1 and Appendix C.1). If a segment contains only advertisements, it is unlikely to be one of the ground-truth highlights, as the labels are aggregated over 50k viewers.
>
> **[R5] Human evaluation of predicted highlights**
>
> A user study about the alignment between human perception and replay peaks has already been done in the literature [1]. To quote their results (Table 5 in [1]), the annotators agreed with 64.8% of the labels. Considering that we even have stricter criteria for highlights, we expect our agreement rate to be higher.
>
> [1] Sul, Jinhwan, et al. Mr. HiSum: A Large-scale Dataset for Video Highlight Detection and Summarization. NeurIPS 2023

---

> > ### Comment · Reviewer_ZtNz · 2025-06-08
> >
> > Thank you for your response.
> > Rgearding, Handling of non-content segments (e.g., ads, intros, boilerplate), have the authors tested this hypotheiss that "LMs should ideally consider those factors and correctly distinguish between extraneous and non-extraneous content"? If not, how such argument can be made?

---

> > ### Author Response · Authors · 2025-06-09
> > **Author Response for Reviewer ZtNz**
> >
> > To answer your question, we have investigated some (transcript segment, segment summary) pairs. To find segments that include advertisements, we searched for the phrase 'sponsored by.' Here, we provide some qualitative examples (*italics* indicate the ads):
> >
> > > **TRANSCRIPT**
> > even just again, on that flexing side is gonna become a thing, but the utility is where the real play is. All right, so that is the synopsis of web one, two, and three. *This episode is sponsored by Athletic Greens. Get a free one-year supply of vitamin D and five free travel packs with your first*
> > >
> > > **SUMMARY**
> > He reiterates that while the visibility of ownership will be valuable, the practical utility of NFTs is of greater significance, summing up the evolution from Web1 to Web3.
> >
> > > **TRANSCRIPT**
> > Thanks for tuning into this episode of Health Theory, *sponsored by our friends at Athletic Greens. We've got an awesome offer for you guys in the description below, so be sure to check that out.* In today's episode with Dr. Ken Berry, we discuss why your doctor is lying to you, the importance of eating ancestrally, why the US nutrition guidelines are absolute garbage, and the efficacy of the carnivore diet. Hey, everybody. Welcome to Health Theory. Today's guest is Dr. Ken
> > >
> > > **SUMMARY**
> > In this introduction to Health Theory, the host teases a conversation with Dr. Ken Berry about the misconceptions surrounding medical advice, the benefits of an ancestral diet, criticisms of US nutrition guidelines, and insights on the carnivore diet.
> >
> > As shown in these examples, if the segment contains both non-extraneous (podcast) and extraneous (ad) content, the extraneous content is correctly removed by summarization.
> >
> > In some cases, we also found that if a segment only consists of ads, usually they are not removed in the summary. However, the summary does correctly detect that it is extraneous non-podcast content (e.g. *The podcast transitions to a sponsorship message...*). For example:
> > > **TRANSCRIPT**
> > *website is sponsored by Squarespace. Listen, in our increasingly online world, the best way to stand out is with a personalized website. And now you're probably at home thinking, but Bailey, Bailey, I don't know how to make a website. I don't know how. First of all, calm down. Worry not, because Squarespace has done the hard work for you. Squarespace, if you don't know,*
> > >
> > > **SUMMARY**
> > The podcast transitions to a sponsorship message from Squarespace, promoting its user-friendly platform for creating personalized websites.
> >
> > While the LLMs we use for summarization already handle this issue to some extent, we think that future work can explore more effective input preprocessing techniques to improve performance further. We will add this discussion to the paper.

---

> ### Comment · Reviewer_ZtNz · 2025-06-10
>
> Thanks for the reponse. I agree with "preprocessing techniques", and as mentioned in my first comments that technique already exists in the literature [1]. I suggest the authors to add this note to discussion/limitation of the work.
>
> [1] Reddy, S., Yu, Y., Pappu, A., Sivaraman, A., Rezapour, R., & Jones, R. (2021, April). Detecting Extraneous Content in Podcasts. In Proceedings of the 16th Conference of the European Chapter of the Association for Computational Linguistics: Main Volume (pp. 1166-1173).

---

### Official Review · Reviewer_7EoE · 2025-05-13

**Rating:** 6
**Confidence:** 3
**Ethics Flag:** 1

**Summary:**

This paper tackles the challenge of highlight detection in podcasts using podcast transcripts, audio features, and episode titles. The authors create a new dataset that maps YouTube videos to replay graphs that highlight the most watched portions of podcasts. They evaluate models like GPT4o, Gemini and, LLama on this longform comprehension task.

**Questions To Authors:**

-- Did the authors consider using visual information as well ? Could some peaks in the replay graph be correlated with events in the visual scene that may be ignored by the current approach ?

-- Did the authors try this task with audio only ? Models like Gemini are capable of predicting timestamps to the best of my knowledge, and those could be used to correlate directly with the replay graph.

-- What happens if peaks occur at the boundary of segments ?

-- Why not use the transcript segments in full rather than their summaries ?

-- It is not clear if authors have the legal permissions to re-distribute the most replayed feature graphs for public consumption. Specific licensing information should be included in the paper.

**Reasons To Accept:**

-- This paper works on an interesting and important problem.

**Reasons To Reject:**

-- The proposed bias correction mechanism could over-correct, i.e., cases where the first frame is indeed a highlight may not be correctly recognized due to the de-biasing employed.

-- How accurate are the V/A/D scores, the transcripts and the VAD scores ? Errors in these processes could artificially inflate the complexity of this task.

-- The current formulation of the task involves multiple automatic steps to predict possibly derived replay graphs. It is possible that with multimodal LLMs like Gemini used in this paper to do this task in a more direct and simple manner. For example, one could use a Gemini model to transcribe only the highlights with timestamps or transcribe the entire audio but add special tags around highlighted segments etc. It is not clear that the proposed formulation is the best one.

---

> ### Author Response · Authors · 2025-06-03
> **Author Response for Reviewer 7EoE**
>
> We sincerely thank the reviewer for their insightful and valuable comments. Please see our responses below:
>
> **[R1] The proposed bias correction mechanism could over-correct.**
>
> Thanks for raising this point. We were also worried about it at first. We clarify that the first segment can still be identified as the highlight after the correction. As shown in Figure 7, the highlight selection rate of the first segment is adjusted similarly to that of other segments after the correction, indicating the effectiveness of our mechanism.
>
> **[R2] Errors in automatic transcription and V/A/D extraction could artificially inflate the complexity of this task.**
>
> We agree that errors in automatic transcription and audio feature extraction might affect the model's performance. However, these errors don’t affect the complexity of this task, since we formulate the task as predicting highlights given only the audio track of a podcast episode (Section 2.1). In Section 3.1, we report the best option of what we’ve explored for input feature processing. Future works can explore better, error-free input feature processing techniques, as well as better prompts, or different models.
>
> **[R3] It is not clear that the proposed formulation is the best one.**
>
> It is certainly possible to solve the proposed task (highlight detection from a long-form podcast audio clip) in many ways, including the ones you propose. We do not claim that our task formulation is the best way to set up the task, but it is a reasonable one. We will discuss alternative formulations (like the one you suggested) in the paper.
>
> **[Q1] Did the authors consider using visual information as well?**
>
> In this work, we focus on whether the current LMs are capable of finding interesting parts given text or audio. As stated in lines 84-86, we filter out videos in which visual information is important. Although we developed our filtering criteria by manual inspection of our dataset, the dataset may include a few videos where visual information matters, given that our dataset contains 13K podcasts.
>
> **[Q2] Did the authors try this task with audio only?**
>
> We used Gemini with both text and audio, which is a superset of using just the audio for the task. Therefore, we expect that the audio-only version will underperform the text+audio version, as it’s harder to extract semantic information from the audio signal.
>
> **[Q3] What happens if peaks occur at the boundary of segments?**
>
> A peak can’t occur at the boundary of segments, since one data point in a replay graph corresponds to a single segment. Please refer to Section 2.1 for detailed information about the replay graph.
>
> **[Q4] Why not use the raw transcript segments rather than their summaries?**
>
> From our initial experiments, we observed that providing raw transcripts to the models results in worse performance. There might be two major reasons for this. First, raw transcripts don’t provide any delimitation between the interlocutors, which is critical for understanding the content based on text. Second, as shown in Table 1, raw transcripts have ~4.7K words in total, and providing transcripts without any preprocessing results in an extremely long prompt, which can be harmful for the model’s performance.
>
> **[Q5] Do the authors have the legal permissions to redistribute the most replayed feature?**
>
> Thank you for pointing out an important issue. As stated in the ethics statement, there have been many datasets and benchmarks that include YouTube features, such as video metadata [1,2], transcript [3,4], uploader-generated chapters [4], and replay graphs [5]. To further respect the rights of creators, we plan to release only features that are necessary for reproduction.
>
> [1] Zellers, Rowan, et al. MERLOT: Multimodal Neural Script Knowledge Models. NeurIPS 2021
>
> [2] Zellers, Rowan, et al. MERLOT Reserve: Neural Script Knowledge through Vision and Language and Sound. CVPR 2022
>
> [3] Han, Seungju, et al. CHAMPAGNE: Learning Real-world Conversation from Large-Scale Web Videos. ICCV 2023
>
> [4] Yang, Antoine, et al. VidChapters-7M: Video Chapters at Scale. NeurIPS 2023
>
> [5] Sul, Jinhwan, et al. Mr. HiSum: A Large-scale Dataset for Video Highlight Detection and Summarization. NeurIPS 2023

---

> > ### Comment · Reviewer_7EoE · 2025-06-08
> >
> > I thank the authors for their responses.

---

### Official Review · Reviewer_csU1 · 2025-05-13

**Rating:** 8
**Confidence:** 4
**Ethics Flag:** 1

**Summary:**

This paper introduces a new dataset of 13K podcasts paired with which segments are highlights for training and evaluating models on highlight prediction. The evaluation covers both LLMs and trained multimodal and unimodal models, showing that current zero-shot LLMs perform relatively poorly but trained models can recover some signal, though there is still substantial room for improvement. Combing multiple modalities also seems to help.

The paper is highly original and introduces a challenging task that benefits from multiple modalities. The evaluation is thorough and represents a solid amount of work for a COLM paper. The dataset will likely be used by later tasks and has nice parallels with video highlight prediction; however, given the lack of work in audio for podcasts, this resource will likely help inspire much future work in the area. The paper is clearly written and well executed.

**Questions To Authors:**

- The only notable weakness for the paper for me is a lack of an error analysis, which might help guide how to improve performance.
  - Podcasts can vary significantly in length so I wonder if there are issues with model performance relative to how long the segments are. With a mean length of ~30 minutes, does the model perform worse on 60 minute podcasts?
  - The authors have a nice note about the issue of the first segment frequently being a highlight. I would be curious if there is additional structure in the podcast itself that might point to varying error rates by segment number

- I would have liked to see if the authors could get a bit more performance from the commercial LLMs. I still think the fine-tuned model is likely going to be better, but I do think tweaking the prompts some could help. For example, the authors note on Line  208 that the zero shot models could not be controlled for how many highlights they create. However, I could imagine a prompt that informs the model about what the median number of highlights are, or suggests a range of highlights. Prompt engineering can be a rabbit hole, so I'm not suggesting that the authors endlessly tweak their design. However, I do think some stronger prompting designs might tell us a bit more about where the gap is in their performances and what aspects need to be improved.

- Since Figures 4 and 5 are pooling episodes with same number of highlights, I would appreciate seeing errors bars on the plots

- Line 256 notes the 100K podcast dataset of Clifton et al. (2020) as the largest but I think this has been superseded by the 1.1M podcasts dataset of [Litterer et al. (2024)](https://arxiv.org/abs/2411.07892) which is on huggingface https://huggingface.co/datasets/blitt/SPoRC

**Reasons To Accept:**

- New challenging task that benefits from text and audio modalities

- New resource for the community

- Thorough evaluation that covers multimodal and unimodal LLMs, both off the shelf and a custom model that is fine-tuned

**Reasons To Reject:**

The paper is very good so I don't have any strong reasons to reject. However, there are a few weaknesses/critiques in the comments box.

---

> ### Author Response · Authors · 2025-06-03
> **Author Response for Reviewer csU1**
>
> We thank the reviewer for their insightful comments and questions. Please see the answers to your questions below:
>
> **[Q1a] How does podcast duration affect model performance?**
>
> For your better understanding, we report the performance of the models depending on the podcast duration: https://ibb.co/TBLYRDfj (hit rate), https://ibb.co/Xxjct3j9 (precision)
>
> In this figure, we grouped podcast episodes by rounding up the duration to the nearest 10 minutes. I.e., podcasts with a duration of up to 10 minutes belong to datapoint at $x=10$, podcasts with a duration of 10 to 20 minutes belong to datapoint at $x=20$, and so on. We do not see a clear trend, most systems perform similarly regardless of the podcast duration.
>
> **[Q1b] Is there any additional structure in the podcast itself that might point to varying error rates by segment number?**
>
> From the left graph in Figure 7, we can observe mainly two tendencies: (1) the first value is significantly higher than the rest, and (2) values at the end (segment 90-100) are a little bit lower than preceding values. We decided not to consider the second tendency, since the trend is way weaker than the first one. However, we agree that investigating the internal structure in replay graphs is an interesting and important future research direction, since it is related to people’s podcast listening behavior.
>
> **[Q2] Stronger prompting designs (e.g., informing the model about what the median number of highlights is)**
>
> As you pointed out, there might be a stronger prompt than the one we used. The reason why we didn’t provide the model with some statistics on $|G|$ is that we were trying to simulate the real-world inference scenario, where the statistics are not available. That being said, we conducted the initial experiments with a few different prompts and decided to include ‘Include a segment only if there is a highly obvious reason it might be replayed, excluding uncertain segments.’ in the prompt after we found this helps the models to output the appropriate number of predicted highlights.
>
> **[Q3] Error bars on Figures 4 and 5**
>
> We agree that the figures would be more informative if there were error bars. We will shade the area between $\pm1$ standard deviation around the mean in the camera-ready version of the paper.
>
> **[Q4] Update on literature**
>
> Thank you for letting us know about the literature that we weren’t aware of. We will revise the writing accordingly in the camera-ready version of the paper.

---

> > ### Comment · Reviewer_csU1 · 2025-06-10
> >
> > Thank you for the clarifications and it's nice to hear that there's some gains to be had with a bit of prompt engineering. I am still very positive about this paper and will keep my score.

---

### Decision · Program_Chairs · 2025-07-07

**Decision:**

Accept

**Comment:**

Summary of the work:
The work introduces the Rhapsody dataset, consisting of 13k audio podcasts based on YouTube podcast videos. The authors propose to use the "most replayed" metadata provided with Youtube videos to assign "highlight segments" automatically, which is proposed as a challenging task (as the authors argue highlights can help listeners navigate podcasts efficiently). Experiments conducted show poor performance of out-of-the-box methods (ZS LLMs, etc.) and that fine-tuning (including models that leverage both audio and text modality) leads to improvements, though the task remains challenging.

Recommendation:
All reviewers lean positive, and I tend to agree. The idea of using "most replayed" signal, and the successful implementation at scale, is a solid contribution to the community. I recommend the authors carefully integrate discussion points in their draft, including:
- Analysis around podcast length vs. performance and genre
- Additional discussion points, including around human annotation.

Additionally, I strongly recommend that the authors carefully consider how to release the dataset thoughtfully. As mentioned by one of the reviewers, it could be unclear whether "most replayed" metadata is allowed to be scraped and redistributed by Google. Also, it might be thoughtful to allow podcast creators to "opt out" from being included in the collection; this option was available for the Spotify podcast dataset (for example).